# Domain-Constrained Distillation of DINOv3 into a Lightweight Foundation Model Toward Point-of-Care Ultrasound

**Md Jaber Al Nahian**[*1] (iD)                                     MDJABERA@UALBERTA.CA
[1] *Faculty of Medicine and Dentistry-Radiology and Diagnostic Imaging Department, University of Alberta, AB, CA*
**Shrimanti Ghosh**[1]                                          SHRIMANT@UALBERTA.CA
**Jacob Jaremko**[1]                                            JJAREMKO@UALBERTA.CA
**Abhilash Hareendranathan**[†1]                                HAREENDR@UALBERTA.CA

**Editors:** Accepted for publication at MIDL 2026

## Abstract

Vision foundation models such as DINOv3 provide powerful representations but are too computationally demanding for point-of-care ultrasound (POCUS), whereas lightweight CNNs remain deployable yet brittle when faced with diverse anatomies and acquisition styles. We bridge this gap with a domain-constrained distillation framework that transfers DINOv3 ViT-B/16 knowledge into a compact ResNet-50, achieving roughly $3.4\times$ compression while preserving the teacher's billion-scale visual priors. Using a large, heterogeneous ultrasound corpus and physics-aware augmentations, the distilled model delivers substantial linear-probe improvements over standard CNN baselines and consistently outperforms the ViT teacher on challenging, heterogeneous datasets. It further offers marked gains in limited-label regimes, reflecting the realities of POCUS workflows where annotated data are scarce. Embedding visualizations show that the distilled encoder forms clearer, anatomy-aware clusters than the teacher, indicating successful alignment to ultrasound structure. Together, these results demonstrate that large-scale natural-image priors can be distilled into a lightweight, generalizable encoder suitable for resource-constrained clinical deployment.

**Keywords:** DINOv3, Distillation, POCUS, Foundation Model, Domain Adaptation.

## 1. Introduction

Large vision foundation models (FMs) such as DINOv3 and related self-supervised ViT encoders achieve strong transfer across many visual tasks by pretraining on billions of natural images (Radford et al., 2021; Oquab et al., 2023; Siméoni et al., 2025; Kirillov et al., 2023). These models are increasingly attractive for medical imaging, where labeled data are scarce and distribution shifts are common. However, most existing FMs require massive computational resources including GPUs and large memory. These resources are often not available in clinical settings. Modern handheld POCUS probes are frequently paired with smartphones or tablets for acquisition and display (EchoNous, 2025; Knight et al., 2023; Jaremko et al., 2023). Deploying large ViT-based foundation models for real-time inference on such resource-constrained mobile hardware can be challenging due to

---

[*] Corresponding Author
[†] Co-corresponding Author

latency and memory/compute overhead, motivating model compression and distillation for point-of-care use (Li et al., 2022).

ViT-based models are powerful, yet their high memory and processing requirements make them challenging to run efficiently on edge devices in real time (Azad et al., 2024). Compact convolutional neural networks (CNNs) are much easier to deploy, but they are often trained on small private ultrasound datasets and are prone to generalization failures across anatomies, scanners, and acquisition protocols (Wu et al., 2024). In practice, clinicians must choose between accurate but impractical models and practical but brittle ones.

We tackle this deployment–performance trade-off by treating FM adaptation as a *knowledge preservation* problem rather than a pure compression problem. Starting from a DINOv3 ViT-B/16 teacher pretrained on 1.7B natural images, we distill its representations into a lightweight ResNet-50 student trained on a curated, large-scale ultrasound corpus of 162,000 unlabeled B-mode images from 35 diverse public datasets. The aim is not only to reduce parameter count, but to transfer both generic visual structure learned at billion scale (edges, shapes, hierarchical abstractions) and ultrasound-specific appearance patterns shaped by B-mode physics and clinical scanning practice. We implement this *domain-constrained* adaptation by supervising the student only through DINOv3 token embeddings, while training on ultrasound-only data with ultrasound-aware augmentations (horizontal flips, moderate zoom, mild blur) that reflect plausible B-mode acquisition changes.

We show that this distillation strategy produces a compact ultrasound foundation model that, in certain settings, is competitive with or occasionally outperforms the heavy ViT teacher in ultrasound segmentation and classification tasks. It also maintains strong performance in low-label regimes. Representation analyses suggest that the distilled model retains useful natural-image structure while forming anatomically meaningful clusters across diverse ultrasound domains. Overall, our results suggest that billion-scale natural-image pretraining can be transferred into a lightweight CNN without sacrificing accuracy, offering a promising step toward foundation models that better align with the computational constraints of POCUS systems. A detailed discussion of related work on vision FMs, medical FMs, and medical distillation is provided in Section 2.

## 2. Related Work

### 2.1. Ultrasound-specific deep learning

Before FM-style pretraining, ultrasound applications primarily relied on task-specific CNNs trained from scratch or initialized from supervised ImageNet weights (Perdios et al., 2018; Zheng et al., 2023; Inan et al., 2024). U-Net variants and ResNet-based encoders have been widely deployed for lesion segmentation, organ boundary detection, and view classification (Ronneberger et al., 2015; Chen et al., 2018). While compact enough for embedded deployment, these models are typically trained on small, single-center datasets and exhibit poor cross-domain generalization.

## 2.2. Vision foundation models and knowledge distillation

Large-scale vision foundation models including CLIP, DINOv2/DINOv3, and Segment Anything (SAM/SAM2) achieve strong zero-shot and transfer performance across classification, detection, and segmentation by pretraining on hundreds of millions to billions of natural images (Radford et al., 2021; Oquab et al., 2023; Siméoni et al., 2025; Kirillov et al., 2023). Knowledge distillation—where a large teacher supervises a smaller student via feature, logit, or attention matching—is widely used to compress such models for edge deployment (Hinton et al., 2015; Romero et al., 2015; Zagoruyko and Komodakis, 2016). Recent work has distilled SAM-like segmenters and DINO-style self-supervised ViTs into compact students, showing that much of the teacher's representational power can be retained in lighter architectures (Zhang et al., 2023; Kang et al., 2023). These approaches, however, are almost exclusively evaluated on natural-image benchmarks.

## 2.3. Foundation models in medical imaging

Medical imaging has rapidly adopted foundation-model pretraining, with adaptations of SAM (e.g., MedSAM, Sam2Rad) and other large encoders improving performance and label efficiency across CT, MRI, X-ray, and histopathology (Ma et al., 2024; Wahd et al., 2025b; Hosseinzadeh Taher et al., 2023; Pai et al., 2025; Shaikovski et al., 2024). For ultrasound specifically, emerging ultrasound foundation models trained across multiple organs and anatomies demonstrate promising transfer to segmentation and classification under limited labels (Megahed et al., 2025; Jiao et al., 2024b; Ma et al., 2025). Most of these models are trained on large but highly heterogeneous and often imbalanced datasets.

## 2.4. Knowledge distillation in medical imaging

Knowledge distillation has been applied in medical imaging to compress large segmentation networks, ensembles, and self-supervised encoders (Qin et al., 2021; Wang et al., 2023; Vray et al., 2024). Existing techniques distill logits, intermediate feature maps, or contrastive representations, sometimes with uncertainty weighting or region-aware losses. Many prior works primarily use supervised task-specific teachers (discarding a wealth of natural-image priors), focus on a single modality or anatomy, or treat domain adaptation as downstream fine-tuning rather than an integral part of the distillation process. In ultrasound, distillation has mainly been used to compress task-specific models rather than to build general-purpose ultrasound foundation models (Dapueto et al., 2024).

## 2.5. Research Gap and Proposed Solution

Existing foundation models and distillation methods do not systematically address how to preserve large-scale natural-image priors while adapting to specialized medical modalities such as ultrasound. In practice, current ultrasound approaches still trade off between heavy ViT-based FMs (high capacity but impractical for point-of-care deployment) and lightweight CNNs (easy to deploy but not generalizable). Motivated by this gap, we propose a domain-constrained, feature-level distillation framework that transfers DINOv3 token representations into a compact ResNet-50 student, using ultrasound-only data and ultrasound-aware augmentations to tailor the encoder for POCUS settings.

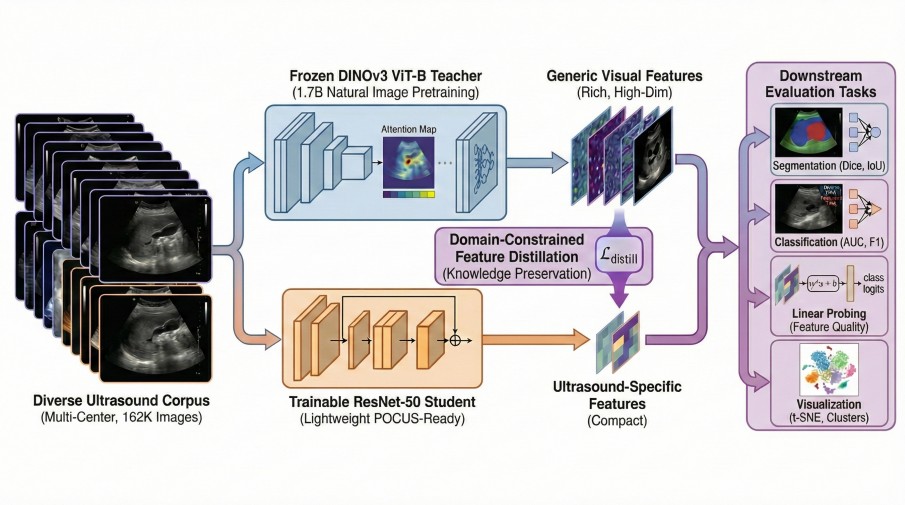

Figure 1: Overview of the proposed domain-constrained distillation framework. A frozen DINOv3 ViT-B/16 teacher produces rich generic visual representations that are distilled into a compact ResNet-50 student via a token-wise feature alignment loss $\mathcal{L}_{\mathrm{distill}}$, using ultrasound-only data and ultrasound-aware augmentations. The resulting encoder is evaluated on downstream segmentation, classification, linear probing, and representation analysis.

## 3. Method

Our goal is to obtain a compact, ultrasound-specific encoder by distilling a large DINOv3 ViT-B/16 teacher into a ResNet-50 student using a large corpus of unlabeled ultrasound images. Fig. 1 summarizes the pipeline. In this section, we describe the ultrasound corpus, the teacher–student architecture, the feature-level distillation objective, the ultrasound-aware augmentations, and the optimization details.

### 3.1. Ultrasound Corpus

We curate a heterogeneous corpus of ∼162,000 unlabeled B-mode ultrasound images spanning diverse anatomies (breast, thyroid, cardiac, fetal, and musculoskeletal) and acquisition settings. The corpus is formed by converting volumetric/video ultrasound data into 2D frames: for 3D studies we export 2D slice frames, and for video clips we extract a subset of informative, non-redundant frames to reduce near-duplicates while preserving clinically meaningful content. Beyond this frame extraction, we apply minimal preprocessing, including basic cleaning (removing overlays/annotations when present) and discarding non-ultrasound or corrupted frames. A detailed list of the constituent datasets and sources is provided in Table 1. No labels are used during training, encouraging the encoder to learn transferable, anatomy-agnostic ultrasound representations.

Table 1: Overview of the public and institutional datasets comprising the ultrasound corpus, organized by anatomy and source.

| Dataset (Ref) | Anatomy | Dataset (Ref) | Anatomy |
|---|---|---|---|
| **105 US Images** (Hann et al., 2017) | Liver | **LUS Phantom** (McLaughlan et al., 2024) | Lung |
| **AbdomenUS** (Vitale et al., 2020) | Abdomen | **MicroSeg** (Shao and Brisbane, 2024) | Prostate |
| **AULI** (Yiming et al., 2022) | Liver | **MMOTU-2D** (Zhao et al., 2022) | Ovary |
| **Brachial Plexus** (Tyagi et al., 2024) | Nerve | **MMOTU-3D** (Zhao et al., 2022) | Ovary |
| **BrEaST** (Pawłowska et al., 2024) | Breast | **PSFHS** (Jieyun and ZhanHong, 2024) | Fetal Head |
| **BUS-UCLM** (Vallez et al., 2025) | Breast | **mu-RegPro** (Baum et al., 2023) | Prostate |
| **BUSBRA** (Gómez-Flores et al., 2024) | Breast | **S1** (Guo et al., 2021) | Breast |
| **BUS_UC** (Iqbal and Sharif, 2024) | Breast | **Segthy** (Krönke et al., 2022) | Thyroid |
| **Cactus Dataset** (Elmekki et al., 2025) | Cardiac | **STMUS** (Marzola et al., 2021) | MSK |
| **CAMUS** (Leclerc et al., 2019) | Cardiac | **STU Hospital** (xbhlk, 2026) | Thyroid |
| **Carotid Artery** (Momot, 2022) | Carotid Artery | **US Fetus** (Anitha, 2024) | Fetus |
| **COVID-BLUES** (Wiedemann et al., 2025) | Lung | **UBPD** (Ding et al., 2022) | Brachial Plexus |
| **EchoCP** (Xu et al., 2020) | Cardiac | **HC** (van den Heuvel et al., 2018) | Fetal Head |
| **EchoNet-Dynamic** (Ouyang et al., 2020) | Cardiac | **JNU-IFM** (Lu et al., 2022) | Fetal Head |
| **Fast-U-Net** (Ashkani Chenarlogh et al., 2022) | Fetal Head | **KidneyUS** (Singla et al., 2023) | Kidney |
| **FASS** (Da et al., 2023) | Fetal Abdomen | **GIST514-DB** (He et al., 2022) | Gastrointestinal |
| **Fetal Echo** (Stoean et al., 2021) | Fetal Heart | **Injury Loc.** (Kumar et al., 2025) | Spinal Cord |
| **Fetal Plane** (Burgos-Artizzu et al., 2020) | Multi-organ | | |

## 3.2. Teacher and student architectures

We adopt a high-capacity DINOv3 vision transformer as the teacher and a compact ResNet as the student.

**Teacher: DINOv3 ViT-B/16.** The teacher encoder $T$ is a DINOv3 ViT-B/16 model pre-trained self-supervised on $\approx$ 1.7 billion natural images. For an input image $x \in \mathbb{R}^{3 \times 224 \times 224}$ (grayscale replicated to three channels), the ViT processes non-overlapping 16×16 pixel patches, yielding a 14×14 grid of $N_{\text{tok}} = 196$ patch token embeddings. Patch token embeddings from the last $n = 2$ transformer blocks are extracted, concatenated along the feature dimension, and flattened in raster-scan order to produce:

$$Z_T(x) \in \mathbb{R}^{196 \times 2d_T}, \tag{1}$$

where $d_T = 768$ is the ViT-B embedding dimension, giving a final teacher representation of dimension $2d_T = 1536$. The teacher is frozen throughout training.

**Student: ResNet-50.** The student encoder $S$ is a standard ResNet-50. Given $x$, the student produces a convolutional feature map

$$F_S(x) \in \mathbb{R}^{C_S \times H_S \times W_S}. \tag{2}$$

This feature map is passed through a projection head $g_\phi$ (a single linear layer) that maps each spatial location to the teacher embedding dimension $2d_T = 1536$, then bilinearly upsampled to match the teacher's 14×14 resolution and flattened in raster-scan order:

$$Z_S(x) = g_\phi\big(F_S(x)\big) \in \mathbb{R}^{196 \times 2d_T}. \tag{3}$$

The ViT-B/16 teacher has $\sim$86M parameters, whereas the ResNet-50 student has $\sim$25M parameters, yielding a $\sim$ 3–4$\times$ reduction in parameter count and model size.

## 3.3. Feature-level distillation

We use a feature-level knowledge distillation scheme that aligns teacher and student token embeddings on unlabeled ultrasound images. For each image $x$, we apply a stochastic

augmentation $a(\cdot)$ to obtain $\tilde{x} = a(x)$ and feed the same view to both teacher and student:

$$Z_T = Z_T(\tilde{x}) = T(\tilde{x}) \in \mathbb{R}^{196 \times 2d_T}, \tag{4}$$

$$Z_S = Z_S(\tilde{x}) = g_\phi\big(F_S(\tilde{x})\big) \in \mathbb{R}^{196 \times 2d_T}. \tag{5}$$

**Token Alignment.** To ensure spatial consistency during distillation, patch token embeddings from the last $n = 2$ transformer blocks of the ViT-B/16 teacher are extracted and spatially resized if necessary to match the last block's 14×14 resolution. They are then concatenated along the feature dimension into a unified representation of shape $(B,\ 2d_T,\ 14,\ 14)$, and flattened in raster-scan order into $(B,\ 196,\ 2d_T)$. The ResNet-50 student's convolutional feature map is projected via a single linear layer and bilinearly upsampled to match the teacher's 14×14 resolution, then similarly flattened in the same raster-scan order into $(B,\ 196,\ 2d_T)$. Both sequences are supervised using a token-wise MSE loss without feature normalization, without modifying either backbone.

**Distillation loss.** The distillation loss is implemented as a token-wise mean squared error (MSE) between teacher and student embeddings without feature normalization. Let $Z_T(i) \in \mathbb{R}^{2d_T}$ and $Z_S(i) \in \mathbb{R}^{2d_T}$ denote the embeddings of the $i$-th token in the sequence, where $i = r \cdot 14 + c$ indexes the spatial location $(r, c)$ in the 14×14 grid. Spatial correspondence between the $i$-th teacher and student tokens is enforced by construction: both representations are reshaped to the same 14×14 grid prior to flattening in raster-scan order, ensuring that token $i$ in both sequences corresponds to the same image region. The loss for one image is

$$\mathcal{L}_{\text{distill}}(x) = \frac{1}{196} \sum_{i=1}^{196} \big\| Z_T(i) - Z_S(i) \big\|_2^2, \tag{6}$$

and the batch loss is obtained by averaging (6) across the mini-batch. We do not use any additional logit-based distillation or contrastive loss; all supervision is mediated through the teacher token embeddings.

**Mixup regularization.** We further apply image-level mixup within each batch as a regularizer. Given two images $x_a$ and $x_b$ and a mixing coefficient $\lambda \sim \mathcal{U}(0, 1)$, the mixed input is

$$\tilde{x}_{\text{mix}} = \lambda \tilde{x}_a + (1 - \lambda)\tilde{x}_b. \tag{7}$$

The mixed image is then forwarded through both teacher and student, so that the resulting token embeddings implicitly reflect the convex combination of the two input images. This encourages smoother transitions in feature space and improves stability during training with large batches.

### 3.4. Ultrasound-aware data augmentations

We restrict distillation to transformations that reflect real ultrasound acquisition. Images are resized and cropped to mimic natural variation in zoom and field-of-view; horizontal flips are allowed, but vertical flips are excluded because ultrasound probes have a fixed orientation relative to the skin surface, making upside-down views physically impossible in clinical practice. Mild Gaussian blur models depth-dependent defocus, and color jitter is

removed because ultrasound is inherently grayscale. These choices ensure that both teacher and student learn invariances tied to actual probe motion and imaging physics, forming a strictly "domain-constrained" augmentation pipeline.

## 3.5. Optimization and implementation details

We distill a ResNet-50 student from a DINOv3 ViT-B/16 teacher using mini-batch training on the unlabeled ultrasound corpus. A lightweight two-layer MLP projects student features into the teacher embedding space, and tokens from two intermediate ViT blocks are used as supervision. Training runs for 1000 epochs with AdamW (learning rate $1 \times 10^{-4}$, weight decay 0.05), batch size 512, and mixed bfloat16 precision on NVIDIA L40 GPUs.

## 4. Experiments

### 4.1. Tasks and datasets

We evaluate the proposed encoder on two segmentation tasks and one classification task. DDTI is a thyroid nodule segmentation dataset of 637 B-mode ultrasound images with expert pixel-level nodule masks (Pedraza et al., 2015). We use 445 images for training, 127 for validation, and 65 for testing, with splits constructed at the patient level to avoid leakage. BUSI (Al-Dhabyani et al., 2020) is a breast ultrasound dataset of 780 images with lesion masks and image-level labels. For BUSI segmentation, we use the provided binary lesion masks for benign, malignant, and normal cases and create train, validation, and test splits analogous to DDTI. For BUSI classification, we define a three-class problem with normal (133 images), benign (437 images), and malignant (210 images), using disjoint class-stratified train, validation, and test sets built from class-wise folders. Importantly, neither DDTI nor BUSI is included in the unlabeled ultrasound corpus used for distillation, ensuring that downstream evaluation is performed on held-out datasets not seen during the distillation stage.

### 4.2. Models and baselines

All downstream experiments use ResNet-50 backbones with three initialization schemes: (i) **R50-Rand**, where weights are randomly initialized at the start of downstream training (no pretraining) and the model is then trained end-to-end using the same downstream protocol as all baselines; (ii) **R50-Distill-Default** (ours), initialized from a ResNet-50 distilled from a DINOv3 ViT-B/16 teacher on the curated ultrasound-only corpus using default ImageNet augmentations; and (iii) **R50-Distill-US** (ours), initialized from the same distillation procedure but using ultrasound-aware augmentations on the same ultrasound-only corpus, as described in Section 3. For segmentation, each ResNet-50 variant is used as the encoder in the same U-Net-style architecture implemented with Segmentation Models PyTorch, so that only the backbone initialization differs. As a high-capacity reference, we also fine-tune a DINOv3 ViT-B/16 backbone with a lightweight segmentation head. For BUSI classification, we use a standard ResNet-50 classifier (global average pooling followed by a linear head) with the three initializations above.

## 4.3. Training Protocols, Implementation Details, and Evaluation Metrics

**Training Protocols.** We evaluate models under: (1) *Linear Probing*, where the encoder is frozen and only a linear head is trained; and (2) *Full Fine-Tuning*, where the backbone and head are trained end-to-end. All decoders/heads were initialized from scratch during fine-tuning. For fairness, all baselines use the same data splits and task-specific training recipe, with a single learning rate for backbone and head (no layer-wise LR decay). All experiments were conducted on a single NVIDIA L40 GPU on the Compute Canada *Vulcan* cluster.

**Implementation Details.** All models (DINOv3 teacher, distilled ResNet-50 student, and downstream baselines) are trained using 3-channel inputs to match standard pretrained backbones. Since ultrasound images are grayscale, we convert each image to a 3-channel tensor by channel-replication (i.e., copying the same intensity map to R/G/B). For segmentation, we fine-tune for 50 epochs at $256\times256$ (batch 32) using AdamW ($lr=10^{-4}$) with cosine warm restarts and loss $0.8\,\mathcal{L}_{\text{Dice}} + 0.2\,\mathcal{L}_{\text{Focal}}$. For classification, we fine-tune for 50 epochs at $224\times224$ (batch 16) using AdamW ($lr=10^{-4}$) with cosine warm restarts and cross-entropy.

**Evaluation Metrics.** Segmentation performance is evaluated using the Mean Dice Similarity Coefficient (DSC) and Mean Intersection over Union (mIoU), computed *per-image* for the foreground class and averaged over the test set. For classification, we report Accuracy and F1-score. To account for class imbalance, the F1-score is computed via *macro-averaging*: precision and recall are calculated for each class $c$ independently and averaged with equal weight ($F1 = \frac{1}{C}\sum_c F1_c$).

## 5. Results and Discussion

### 5.1. Linear probing on frozen encoders

Table 2 summarizes linear-probe performance across segmentation and classification tasks. On DDTI, both distilled models substantially outperform the randomly initialized baseline, with **R50-Distill-Default** obtaining the best Dice (0.7378) and IoU (0.6252), indicating successful transfer of ViT teacher knowledge to a compact CNN. In contrast, the **DINOv3 ViT-B/16** teacher underperforms (Dice 0.6503), reflecting limited robustness to thyroid-domain grayscale and speckle statistics. On BUSI, the domain gap becomes more pronounced: the teacher collapses to a Dice of 0.2384, while the ultrasound-aware **R50-Distill-US** achieves the strongest segmentation performance (Dice 0.6083, IoU 0.5259). Similarly, **R50-Distill-US** yields the best BUSI classification accuracy (0.7452), whereas **R50-Distill-Default** attains the highest macro F1 (0.7037). These results confirm that distillation on ultrasound-only data, paired with physics-consistent augmentations, produces representations substantially better aligned with downstream ultrasound tasks than the generic natural-image ViT teacher.

### 5.2. Full fine-tuning on all labeled data

As shown in Table 3, full fine-tuning amplifies the benefits of distillation. Both distilled ResNet-50 models consistently outperform the randomly initialized baseline across all tasks. On DDTI, the ultrasound-aware student achieves near-parity with the ViT teacher despite

Table 2: Linear probing on frozen encoders. DDTI and BUSI segmentation are evaluated by mean Dice and mean IoU. BUSI classification is a 3-way task (normal/benign/malignant) evaluated by accuracy and macro F1.

| Model | DDTI Seg. | | BUSI Seg. | | BUSI Cls. | |
|---|---|---|---|---|---|---|
| | Dice | IoU | Dice | IoU | Acc | F1 |
| R50-Rand | 0.6028 | 0.4647 | 0.4365 | 0.3369 | 0.6115 | 0.4204 |
| R50-Distill-Default | **0.7378** | **0.6252** | 0.5771 | 0.4857 | 0.7325 | **0.7037** |
| R50-Distill-US | 0.7334 | 0.6192 | **0.6083** | **0.5259** | **0.7452** | 0.6768 |
| DINOv3 ViT-B/16 | 0.6503 | 0.4743 | 0.2384 | 0.1729 | – | – |

Table 3: Full fine-tuning on all labeled data. Metrics as in Table 2.

| Model | DDTI Seg. | | BUSI Seg. | | BUSI Cls. | |
|---|---|---|---|---|---|---|
| | Dice | IoU | Dice | IoU | Acc | F1 |
| R50-Rand | 0.6605 | 0.5297 | 0.5525 | 0.4629 | 0.6561 | 0.6526 |
| R50-Distill-Default | 0.7652 | 0.6608 | **0.6967** | **0.6209** | 0.8662 | 0.8572 |
| R50-Distill-US | 0.7872 | **0.6745** | 0.6930 | 0.6126 | **0.8790** | **0.8673** |
| DINOv3 ViT-B/16 | **0.7933** | 0.4790 | 0.2838 | 0.1957 | – | – |

using a fraction of the parameters, indicating that distillation preserves performance while substantially reducing model size.

A sharper contrast emerges on BUSI. While DINOv3 adapts well to DDTI, it transfers poorly to BUSI and exhibits unstable fine-tuning behavior: training loss decreases while validation loss degrades and becomes unstable (Supplementary Fig. A3), consistent with overfitting under limited and heterogeneous BUSI supervision (best-validation checkpoint reported). We further analyze this DDTI–BUSI gap in the Supplementary material (Supplementary Sec. A; Supplementary Figs. A2–A1), documenting frequent boundary mismatch, missed small/thin lesions, and false positives on negative (normal-class) images. In contrast, both distilled students fine-tune reliably on BUSI and achieve substantially higher segmentation Dice/IoU and classification accuracy/F1 (Table 3).

Collectively, these results suggest that domain-constrained distillation provides dual benefits: (i) model compression that improves deployability on resource-constrained medical devices, and (ii) an ultrasound-adapted initialization that mitigates natural-image transfer bias and improves robustness on challenging breast ultrasound data.

### 5.3. Limited-label regimes

As shown in Table 4, both distilled models maintain strong performance even when fine-tuned with only a small fraction of labeled data, whereas the randomly initialized baseline degrades quickly. The ultrasound-aware student is particularly stable in the lowest-label settings, indicating that domain-constrained distillation yields features that transfer more reliably under scarce supervision. These trends are most evident in BUSI classification, where the distilled encoder consistently outperforms alternatives across all label fractions. This highlights a key advantage of our approach: by embedding ultrasound-specific priors during distillation, the model becomes far less dependent on large annotated datasets. Such label efficiency is essential for point-of-care and resource-limited environments, where expert annotation is costly or unavailable.

Table 4: Limited-data performance for different label fractions. We report segmentation Dice on DDTI (thyroid) and BUSI (breast), and BUSI 3-way classification accuracy, when fine-tuning on 5%, 10%, 20%, and 50% of labeled data.

| Model | DDTI Dice | | | | BUSI Dice | | | | BUSI Acc | | | |
|---|---|---|---|---|---|---|---|---|---|---|---|---|
| | 5% | 10% | 20% | 50% | 5% | 10% | 20% | 50% | 5% | 10% | 20% | 50% |
| R50-Rand | 0.4269 | 0.5237 | 0.5597 | 0.6007 | 0.1911 | 0.3381 | 0.3895 | 0.4038 | 0.2675 | 0.5605 | 0.6242 | 0.6561 |
| R50-Distill-Default | 0.4958 | 0.6536 | 0.7109 | **0.7672** | 0.2494 | **0.4857** | 0.5011 | **0.6185** | 0.4968 | 0.5860 | 0.6688 | 0.7452 |
| R50-Distill-US | **0.5233** | **0.6574** | **0.7309** | 0.7661 | **0.2752** | 0.4600 | **0.5036** | 0.5543 | **0.7197** | **0.6242** | **0.8025** | **0.8726** |

Table 5: Deployment-oriented comparison of teacher vs. student

| Metric | Teacher (DINOv3 ViT-B) | Student (ResNet50) | Student vs Teacher |
|---|---|---|---|
| Parameters (backbone only) (M) ↓ | 86 | 25 | 3.4× smaller |
| Parameters (backbone + head) (M) ↓ | 87.44 | 32.52 | 2.69× smaller (−62.81%) |
| FLOPs (GFLOPs) ↓ | 22.80 | 10.67 | 2.14× less (−53.18%) |
| GPU latency (ms, mean±std) ↓ | 8.63 ± 0.91 | 4.07 ± 1.26 | 2.12× faster |
| GPU throughput (img/s) ↑ | 86.21 | 174.88 | 2.03× higher |
| Peak GPU memory (MB) ↓ | 686.70 | 372.17 | 1.85× less (−45.80%) |
| CPU latency (ms, mean±std) ↓ | 430.51 ± 3.02 | 33.76 ± 4.50 | 12.75× faster |
| CPU throughput (img/s) ↑ | 2.30 | 27.64 | 12.02× higher |
| **Dice Score** | **0.7933** | **0.7872** | **-0.0061 (-0.77%)** |

## 5.4. Deployment Feasibility for Point-of-Care Ultrasound

Point-of-care ultrasound (POCUS) systems are often deployed on resource-constrained tablet platforms (e.g., EchoNous Kosmos on iPad Air and Philips Lumify on Android tablets), where available system memory is typically on the order of 4–12 GB. (EchoNous, 2025; Knight et al., 2023; Jaremko et al., 2023) To quantify deployment feasibility, Table 5 reports latency, throughput, peak memory, FLOPs, and parameter counts under a standardized protocol (batch size 1, 224×224 input, FP32; CPU timing averaged over 50 runs on an Intel CPU with 8 GB RAM; peak memory measured as maximum GPU allocation on an NVIDIA L40). Compared to the teacher, the distilled student reduces peak GPU memory from 686 MB to 372 MB (45.8%) and achieves substantially faster CPU inference, with latency improving from 430 ms to 33.76 ms per image (12.75×) and throughput from 2.30 to 27.64 images/s (12.02×). The student further reduces FLOPs by 53.18% and parameters by 62.81%, while retaining 99.2% of the teacher's Dice (0.7933 → 0.7872). Together, these results demonstrate a favorable accuracy–efficiency trade-off and support practical deployability under POCUS hardware constraints; on-device benchmarking on commercial scanners is left for future work.

## 5.5. Representation analysis

t-SNE projections of the ultrasound corpus (Fig. 2) show clear differences in how each model organizes the data. The randomly initialized encoder produces highly entangled embeddings with little anatomical separation, consistent with its weaker downstream performance. Distillation markedly improves structure: the default student forms more coherent clusters, while the ultrasound-aware student produces the most distinct and stable separation across datasets. In contrast, the ViT teacher retains broad natural-image structure but shows substantial overlap between ultrasound domains, mirroring its poor BUSI performance. These patterns support our central claim that domain-constrained distillation realigns feature

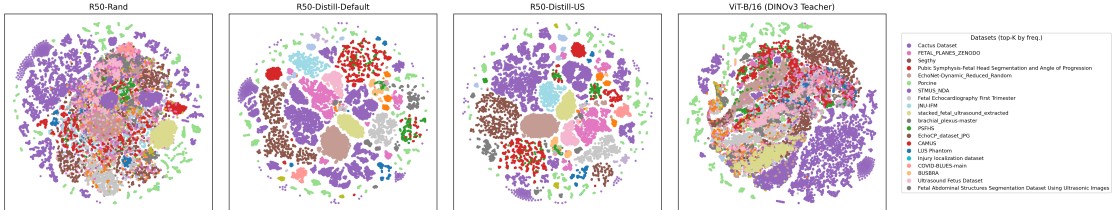

Figure 2: t-SNE visualization of Ultrasound corpus embeddings for four models: R50-Rand, R50-Distill-Default, R50-Distill-US, and ViT-B/16 (DINOv3 Teacher). Points are colored by dataset, with a shared color map across models; the right-most panel shows the legend for the top-$K$ most frequent datasets. R50-Distill-US forms the most compact and well-separated clusters across ultrasound domains.

space toward ultrasound-specific cues, enabling stronger generalization on heterogeneous clinical data.

Table 6: Ablation: Augmentation Policy & Mixup on BUSI Classification & DDTI Segmentation.

| Augmentation | Mixup | BUSI Acc. | F1 | DDTI Dice | DDTI IoU |
|---|---|---|---|---|---|
| Default | ✗ | 0.8662 | 0.8572 | 0.7652 | 0.6608 |
| Default | ✓ | 0.8712 | 0.8602 | 0.7732 | 0.6658 |
| US-aware | ✗ | 0.8790 | 0.8738 | 0.7759 | 0.6705 |
| US-aware | ✓ | **0.8790** | **0.8673** | **0.7872** | **0.6745** |

### 5.6. Ablation Study on Augmentation Policy & Mixup

Table 6 examines the effects of the augmentation policy and mixup on BUSI classification and DDTI segmentation; mixup is applied only during distillation and is not used during downstream fine-tuning or evaluation. The Default augmentation pipeline includes standard ImageNet-style transformations such as random crop, horizontal flip, color jitter, and Gaussian blur. The US-aware policy consistently outperforms Default + Mixup. While mixup improves BUSI accuracy and DDTI Dice slightly when applied to the Default augmentation, it does not significantly boost performance compared to the US-aware augmentation. This suggests that US-aware augmentations are the primary driver of performance improvements, and mixup acts as a regularizer rather than a core component of the framework. Removing mixup under the US-aware setting results in slightly better F1 scores, confirming that ultrasound-aware augmentations alone are sufficient for strong generalization.

### 5.7. Ablation Study on Different Distilled Architectures

We applied the DINOv3 ViT-B16 teacher, US corpus, and US-aware augmentations to distill ResNet-18 and ConvNeXt-Tiny, alongside ResNet-50. Table 7 compares their performance with randomly initialized baselines. All distilled models show significant improvements in BUSI classification and DDTI segmentation. ResNet-18 achieves 9.6% higher BUSI accuracy and 8.9% higher DDTI Dice, while ConvNeXt-Tiny shows a 60.0% increase in BUSI

Table 7: Performance comparison of additional distilled models (ResNet-18 and ConvNeXt-Tiny) versus random initialization.

| Model | BUSI Acc | BUSI F1 | DDTI Dice | DDTI IoU |
|---|---|---|---|---|
| R18-Rand | 0.5987 | 0.4144 | 0.6936 | 0.5657 |
| **R18-Distill-US** | 0.6561 | 0.4835 | 0.7552 | 0.6424 |
| ConvNeXt-Tiny-Rand | 0.5096 | 0.3386 | 0.6251 | 0.4828 |
| **ConvNeXt-Tiny-Distill-US** | 0.8153 | 0.7855 | 0.7543 | 0.6386 |
| **R50-Distill-US** | 0.8790 | 0.8673 | 0.7872 | 0.6745 |

Table 8: Comparison against existing foundation models (FMs) on **DDTI** segmentation after full fine-tuning

| Model | FM type | Backbone | Params (M) | Mean Dice |
|---|---|---|---|---|
| DINOv3 ViT-B/16 | Natural-image FM (SSL) | ViT-B/16 | 86.0 | 0.7933 |
| Temporal (SAM2) | SAM2-based | Hiera-B+ | 80.8 | **0.8041** |
| USFM | Ultrasound-specific FM | ViT-B/16 | 86 | 0.5760 |
| R50-Distill-US (Ours) | Distilled compact FM (US corpus) | ResNet-50 | **25.0** | 0.7872 |

accuracy and 20.6% improvement in DDTI Dice, validating the effectiveness of domain-constrained distillation. While ResNet-50 remains the top performer, these results confirm the framework's generalizability across architectures.

## 5.8. Comparison against existing foundation models (FMs)

Table 8 compares our model with representative FMs after full fine-tuning on DDTI under the same protocol. Temporal (SAM2/Hiera-B+) (Wahd et al., 2025a) performs best (Dice = 0.8041), while our R50-Distill-US is close (Dice = 0.7872) with far fewer parameters (25M vs. 80.8M–86M). Our distilled ResNet-50 also matches the natural-image ViT FM baseline (DINOv3, Dice = 0.7933), supporting our goal of achieving FM-level accuracy in a compact backbone. ViT-based ultrasound specific FM, USFM (Jiao et al., 2024a) performs substantially lower (Dice = 0.5760) on this benchmark.

## 6. Conclusion

We introduced a domain-constrained distillation framework that transfers billion-scale ViT representations into a compact ResNet-50 suitable for ultrasound. Using a curated ultrasound corpus and ultrasound-aware augmentations, the distilled models offer stronger generalization and substantially better label efficiency than both a randomly initialized CNN and the original DINOv3 ViT teacher. These findings demonstrate that large vision priors can be preserved in a deployment-friendly backbone while mitigating the natural-image biases that limit direct ViT adaptation. Our approach provides a simple and practical recipe for building ultrasound foundation models that are compatible with point-of-care constraints. Limitations include the focus on 2D B-mode data and the absence of on-device latency evaluation. Future work will extend this framework to video-based POCUS, additional anatomies, and hardware-aware model design.

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

## Appendix A. Supplementary: Why DINOv3 Behaves Differently on DDTI vs. BUSI

Figures SA2 and SA1 show that DINOv3 transfers more reliably to DDTI than to BUSI under the same full fine-tuning protocol. On DDTI, predictions are spatially localized and largely follow the annotated nodule boundaries, with occasional errors mainly on very small or low-contrast nodules (Fig. SA2, Case 5). In contrast, BUSI exhibits diverse failure modes, including boundary mismatch/over-segmentation (Cases 1–2), missed small/thin lesions (Cases 3–4), and false positives on *normal-class* images where no lesion is annotated (Cases 5–6 in Fig. SA1).

These qualitative patterns are consistent with three dataset-specific factors that make BUSI harder for direct transfer: (i) larger lesion-scale variability and more ambiguous boundaries; (ii) explicit inclusion of negative (normal) cases, increasing false-positive risk; and (iii) higher acquisition and speckle/background variability, which likely amplifies domain shift for a high-capacity natural-image ViT. We further observe unstable optimization on BUSI (Fig. SA3), where training loss continues to decrease while validation loss degrades, suggesting overfitting under standard full fine-tuning. All reported DINOv3 results use the checkpoint selected by best validation performance (same selection rule for all models). Together, these observations motivate our domain-constrained distillation strategy to obtain a compact ultrasound-adapted initialization that fine-tunes more robustly on BUSI while remaining competitive on DDTI.

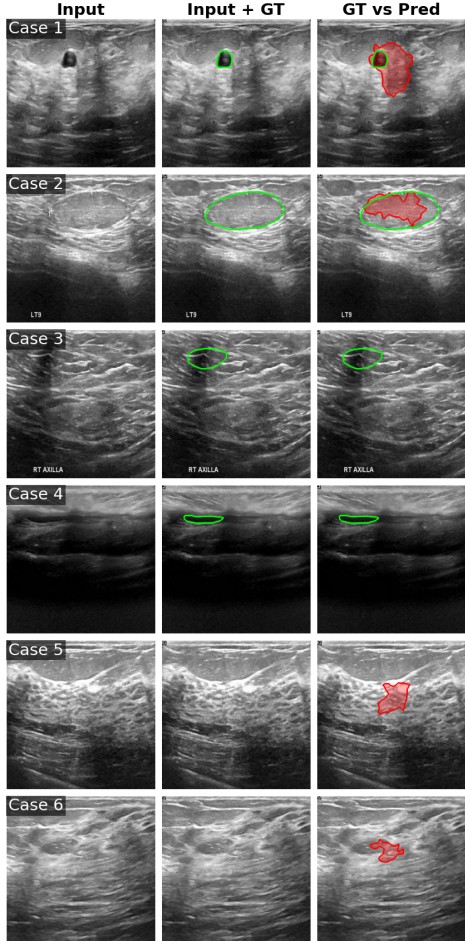

Figure A1: **Qualitative BUSI segmentation failure cases for fine-tuned DINOv3.** Columns show *Input, Input + GT* (ground-truth lesion contour in green when available), and *GT vs Pred* (DINOv3 prediction in red overlaid with GT in green). **Case 1–2**: over-segmentation and boundary mismatch (prediction extends beyond the annotated lesion). **Case 3–4**: false negatives, including missed and thin/small targets (GT present but prediction absent). **Case 5–6**: false positives on *normal-class* images (no lesion annotation in GT), where the model predicts spurious regions.

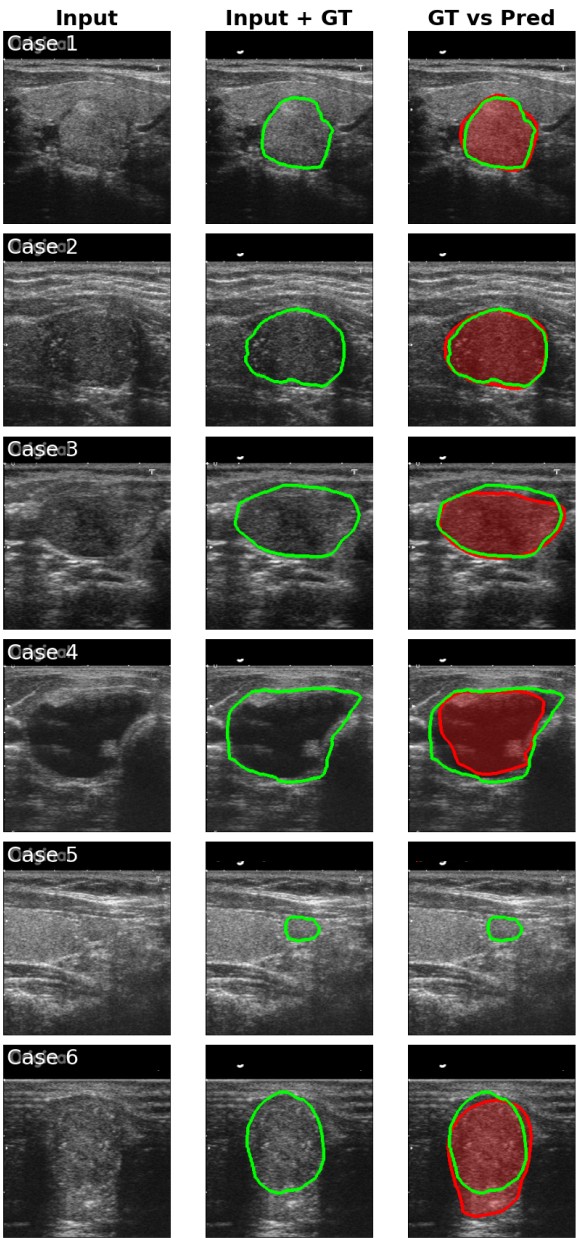

Figure A2: **Qualitative DDTI segmentation results for fine-tuned DINOv3.** Columns show *Input*, *Input + GT* (ground-truth nodule contour in green), and *GT vs Pred* (DINOv3 prediction in red overlaid with GT in green). Across representative cases, predictions are spatially focused and closely match the annotated nodule boundaries, with only minor contour deviations. Case 5 illustrates a small-target false negative / under-segmentation: the nodule is small and low-contrast, and DINOv3 produces a weak or nearly absent prediction, missing most of the annotated region. This contrasts with BUSI, where DINOv3 produces frequent false negatives on small lesions and false positives on normal-class images (Figure. SA1).

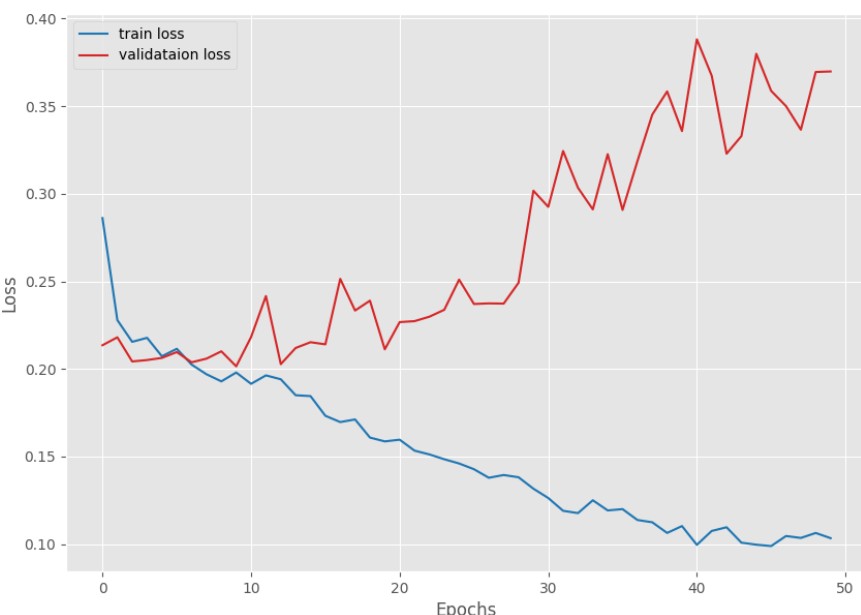

Figure A3: **Optimization behavior of fine-tuned DINOv3 on BUSI.** Training loss decreases steadily over 50 epochs, while validation loss increases and becomes unstable after early training, indicating overfitting during full fine-tuning on BUSI under our standard protocol.

