# OpenReview forum: "Domain-Constrained Distillation of DINOv3 into a Lightweight Foundation Model Toward Point-of-Care Ultrasound"
_MIDL.io/2026/Conference — MIDL 2026 Poster_

### Official Review · Reviewer_9AHR · 2026-01-08

**Confidence:** 3
**Preliminary Rating:** 2
**Final Rating:** 4

**Summary:**

In this paper, the author proposes domain constrained feature distillation to transfer from a frozen DINOv3 into a ResNet-50 for point-of-care-ultrasound (POCUS). The author suggests that with this distilled model, they could achieve better performance while not using as much computational power and memory.

**Strengths:**

- Very clear argument and well organized paper with clear points
- Methods are clearly described and great overview figure
- Experiments showing clear empirical improvements of proposed method over comparing methods or models

**Weaknesses:**

- Please provide more detail about the data and preprocessing. Also reference any data or preprocessing used.
- I can see there is novelty in applying vision distillation for ultrasound dataset, but still a pretty common idea of distillation. Might need more thorough experiments to make a better paper
- I would like authors to provide more experiments on different distilled models other than ResNet-50, and see how performance changes.
- Even though the author's motivation for this distillation model is on computational power and memory limits, not detailed latency/memory measurement such as FLOPs
- I don’t know if you could run DINOv3, but if it is possible, I would like to see what is fine-tuned DINOv3 on the same data results.

**Detailed Comments:**

- Please clarify corpus composition. List all the datasets and accordingly reference them.
- Optionally, it might be a great idea to train and test with other models and ViT or any other ultrasound FM models.
- Can you add more details of evaluation metrics and how they have computed? Is it per-class? per-images? Etc.
- Please provide more details of implementation. For example Dinov3 takes 3 channel input, but ultrasound is greyscale. How did you deal with those kind of details? Please add critical details for your method

**Justification Of Final Rating:**

Authors successfully delivered all of my concerns, and I am changing my rating to 4. Thanks to authors to deliver heavy experiments in such as short amount of time. I believe paper is now a lot stronger and worth to be published in MIDL.

**Justification Of The Preliminary Rating:**

This work provides a straightforward method of ultrasound feature distillation from DINOv3 to the ResNet-50 model. The author suggests this should increase the performance significantly while saving some computational power and memory uses. However, this paper might require more experiments to clearly prove its arguments. The author may add more details of methods, and experiments for higher rating.

**Questions To Address In The Rebuttal:**

I would like to see authors address some of the issues I mentioned on weakness and detailed comments.

---

> ### Author Response · Authors · 2026-01-25
>
> Thank you for the careful review and for pointing out areas where additional experiments and implementation details would strengthen the paper. We made substantial revisions accordingly.
>
> (1) More detail about corpus composition and preprocessing
> We clarified the corpus construction in Sec. 3.1, including how volumetric/video ultrasound is converted into 2D frames, removal of near-duplicates, and minimal preprocessing (e.g., removing overlays/annotations where present, discarding corrupted frames). We also provide a dataset-level list of constituent sources with anatomy labels and references in Table 1.
>
>
> (2) Distillation novelty and need for more experiments
> To strengthen the empirical contribution, we added multiple new experiments/ablations:
>
> Table 6 + Sec. 5.6: Ablation isolating ultrasound-aware augmentation vs mixup.
>
> Table 7 + Sec. 5.7: Distillation applied to additional compact backbones (ResNet-18, ConvNeXt-Tiny), showing consistent gains over random initialization and demonstrating generality beyond ResNet-50.
>
> Table 8 + Sec. 5.8: Comparison against representative foundation model baselines on DDTI segmentation after full fine-tuning.
>
> (3) Deployment evidence (latency/memory/FLOPs)
> We added Table 5 and a new Sec. 5.4 reporting parameters (backbone-only and backbone+head), FLOPs, GPU latency/throughput, peak GPU memory, and CPU latency/throughput under a standardized protocol. This directly supports the claim that distillation improves deployability while maintaining accuracy.
>
> (4) Fine-tuned DINOv3 baseline
> We confirm we fine-tune DINOv3 ViT-B/16 under the same protocol and report results in Tables 2–3 (linear probing and full fine-tuning). We also provide a deeper analysis of why DINOv3 behaves differently on DDTI vs BUSI in Appendix A with qualitative results and an optimization stability plot (Figs. A1–A3).
>
> (5) Evaluation Metrics:
> Section 4.3 clarifies: "Dice/IoU computed per-image for the foreground class and averaged over the test set. F1-score computed via macro-averaging: precision and recall calculated for each class c independently and averaged with equal weight."
>
> (6) Implementation details: grayscale vs 3-channel input
> We added explicit clarification in Sec. 4.3: ultrasound images are grayscale, so we convert them to 3-channel by channel replication to match pretrained backbone interfaces.
>
>
> We appreciate the reviewer’s suggestions; we believe the new ablations, added baselines, and deployment measurements significantly strengthen the manuscript.

---

### Official Review · Reviewer_pBAP · 2026-01-09

**Confidence:** 4
**Preliminary Rating:** 4
**Final Rating:** 4

**Summary:**

This paper studies domain-constrained knowledge distillation to transfer representations learned from natural images (via a strong foundation model teacher DINOv3) to point-of-care ultrasound (POCUS) tasks. The authors propose constraining the distillation process to better respect ultrasound domain characteristics, and they evaluate on multiple POCUS datasets and tasks (e.g., segmentation and classification). Overall, the experiments suggest that distillation can significantly improve downstream performance and robustness compared to directly using (or fine-tuning) the natural-image teacher on ultrasound, supporting the claim that distillation is an effective bridge between natural-image pretraining and POCUS deployment.

**Strengths:**

* Clear motivation and practical relevance: transferring pretrained vision knowledge to POCUS is impactful given limited labeled ultrasound data.

* Empirical evidence of effectiveness: results generally support that distillation improves downstream performance compared with naive transfer.

* Multiple tasks/datasets: evaluation across more than one setting helps demonstrate generality rather than a single benchmark win.

**Weaknesses:**

* Some key claims (especially around “ultrasound-aware augmentation”) are not cleanly isolated from other training choices (e.g., mixup), making it hard to attribute gains.

* The paper reports large task-dependent differences (e.g., DINOv3 behaves very differently across datasets), but the manuscript provides limited mechanistic insight.

* The fine-tuning protocol is not fully specified for all baselines (notably whether the full backbone is updated or partially frozen), which affects reproducibility and fairness.

**Detailed Comments:**

### 1) Mixup vs “ultrasound-aware augmentation”: attribution and ablation
The paper introduces image-level mixup for smoother feature transitions while also arguing that ultrasound-aware augmentations are more reasonable for POCUS. However, mixup itself is not ultrasound-aware in a physical/acquisition sense. As written, it is difficult to tell whether reported improvements are driven by (i) the ultrasound-aware augmentation policy, (ii) mixup regularization, or (iii) their interaction.

### 2) Why does DINOv3 perform so differently across DDTI vs BUSI segmentation?
The manuscript shows that the natural-image foundation model baseline can behave dramatically differently across datasets/tasks (e.g., DDTI vs BUSI segmentation). The paper would be strengthened by a clearer explanation of why this happens. It is currently unclear whether the gap is driven primarily by:
* segmentation target size / lesion scale distribution
* annotation style/noise
* class composition (e.g., inclusion of normal cases)
* general distribution shift (dataset acquisition differences and speckle characteristics)

### 3) Fine-tuning protocol: what is updated (full backbone vs partial)?
For downstream fine-tuning, the paper should clearly state whether the entire teacher/student backbone is updated or staged/partial unfreezing strategy is used. Given limited data in the two PUCUS datasets, finetuning the whole network may not be sufficient.

**Justification Of Final Rating:**

The authors have provided sufficient evidence in their extended experiments to show the effectiveness of the proposed method and fully addressed my concerns. I think the current manuscript is solid enough for more insight discussion in the conference.

**Justification Of The Preliminary Rating:**

The paper presents a convincing empirical case that domain-constrained distillation helps transfer natural-image knowledge to POCUS. With clearer attribution (mixup vs ultrasound-aware augmentation), deeper insight into dataset-dependent behavior (DDTI vs BUSI), and explicit fine-tuning details for reproducibility, the work would be substantially stronger and easier for the community to adopt.

**Questions To Address In The Rebuttal:**

**Augmentation vs mixup (attribution):**
* What exactly is included in the “default” augmentation pipeline?
* Can the authors clarify whether mixup is applied in all settings or only in specific experiments?
* Do the authors have (or can they provide) an ablation isolating the effect of ultrasound-aware augmentation from mixup (e.g., ±mixup under each augmentation policy)?

**DINOv3 discrepancy between DDTI and BUSI segmentation:**
* What do the authors believe is the dominant cause of the large performance gap (e.g., lesion size distribution, acquisition settings, annotation style, class composition, preprocessing differences)?
* Can the authors provide supporting evidence, such as (1) performance stratified by lesion/mask size, (2) qualitative failure cases on BUSI (teacher vs distilled/student), (3) basic dataset statistics or a short diagnostic analysis?

**Fine-tuning protocol clarity and fairness:**
* For each baseline (especially DINOv3), is fine-tuning full-backbone or head-only, or partially unfrozen?
* Are different learning rates used for backbone vs head? Is layer-wise LR decay used for ViT backbones?
* Are training schedules matched across methods to ensure a fair comparison?

**Reproducibility details:**
* Could the authors confirm the key hyperparameters that most affect results (epochs, batch size, LR, weight decay, augmentation strength, distillation temperature / loss weights)?
* Is the code/config planned for release, or can the authors provide sufficient configuration details in the supplement?

---

> ### Author Response · Authors · 2026-01-25
>
> We thank Reviewer pBAP for the detailed and insightful feedback. We address each point:
>
> **Attribution: Mixup vs. US-aware Augmentation:**
> We added **Table 6** with a full 2×2 ablation (±Mixup × Default/US-aware). Key findings:
> - US-aware without mixup: BUSI Acc 0.8790, F1 0.8738
> - Default with mixup: BUSI Acc 0.8712, F1 0.8602
> - US-aware consistently outperforms Default+Mixup
>
> Section 5.6 clarifies: "The Default augmentation pipeline includes standard ImageNet-style transformations such as random crop, horizontal flip, color jitter, and Gaussian blur." Mixup is applied only during distillation. Results confirm US-aware augmentations are the primary driver; "removing mixup under the US-aware setting results in slightly better F1 scores."
>
> **DINOv3 DDTI vs. BUSI Discrepancy:**
> We provide detailed analysis in **Supplementary Section A** with qualitative failure cases (Figs. A1-A3). Three dataset-specific factors make BUSI harder:
> 1. "Larger lesion-scale variability and more ambiguous boundaries"
> 2. "Explicit inclusion of negative (normal) cases, increasing false-positive risk"
> 3. "Higher acquisition and speckle/background variability"
> Fig. A3 shows DINOv3 overfits on BUSI (training loss decreases while validation loss increases). Our distilled students fine-tune more reliably.
>
> **Fine-tuning Protocol Clarity:**
> **Section 4.3** now states explicitly:
> - Linear probing: encoder frozen, only linear head trained
> - Full fine-tuning: backbone and head trained end-to-end
> - "All decoders/heads were initialized from scratch during fine-tuning"
> - "Single learning rate for backbone and head (no layer-wise LR decay)"
> - All experiments on NVIDIA L40 GPU, Compute Canada Vulcan cluster
>
> **Reproducibility:**
> Section 4.3 details: segmentation (50 epochs, 256×256, batch 32, AdamW lr=10⁻⁴, 0.8L_Dice+0.2L_Focal); classification (50 epochs, 224×224, batch 16, cross-entropy). Code and weights will be released upon acceptance.

---

> > ### Comment · Reviewer_pBAP · 2026-01-30
> >
> > Thank you for your response. The overall manuscript looks complete.

---

> > > ### Author Response · Authors · 2026-01-30
> > >
> > > Thank you for your follow-up comment. We appreciate your time and careful review.

---

### Official Review · Reviewer_rGok · 2026-01-09

**Confidence:** 4
**Preliminary Rating:** 4
**Final Rating:** 4

**Summary:**

This paper proposes a domain-constrained distillation framework that transfers representations from a DINOv3 ViT-B/16 into a lightweight ResNet-50 for point-of-care ultrasound applications. By leveraging a large heterogeneous ultrasound dataset and physics-aware augmentations, the distilled model achieves roughly 3.4× compression while preserving the teacher’s billion-scale visual priors.

**Strengths:**

This work addresses an interesting application problem: reducing the computational burden of deploying foundation models on edge devices. By distilling DINOv3 ViT-B/16 into a ResNet-50, the approach substantially reduces model size while largely preserving the representational capacity of the original model.

**Weaknesses:**

1. While the overall direction is interesting and potentially impactful, the reviewer has concerns about the specific application framing of distilling DINOv3 for POCUS. It is unclear whether model size and memory usage during inference truly constitute a bottleneck in this setting, given that inference memory requirements are typically much lower than training. The paper claims that “ViT-based models are powerful, yet their high memory and processing requirements make them challenging to run efficiently on POCUS devices,” but the cited reference (Saha and Xu, 2025) does not explicitly discuss DINO, POCUS, or concrete deployment limitations. Moreover, although the distilled model reduces parameter count from 86M to 25M, both models remain in the same order of magnitude, raising the question of whether this reduction meaningfully addresses real POCUS constraints.

2. Beyond reporting parameter count reduction, the paper does not clearly demonstrate the concrete benefits of distillation in terms of inference speed, memory footprint, or efficiency. In addition, hardware details (e.g., GPU type, batch sizes) used during distillation and evaluation are insufficiently described.

3. Image-level mixup is not physically realistic for ultrasound imaging and may introduce domain mismatch by linearly blending unrelated anatomical structures. While such regularization can be justified in representation distillation settings, its implications in the ultrasound domain are not sufficiently discussed.

4. The experimental section lacks clarity regarding data usage. In Section 3.1, the paper describes a heterogeneous corpus of approximately 160,000 unlabeled ultrasound images but provides limited details about data composition and preprocessing. Later, Section 4.1 introduces DDTI and BUSI for downstream tasks, but it is unclear whether these datasets are also used during DINO distillation or only for downstream training and evaluation.

5. In Section 4.2, the paper introduces “R50-Rand, a ResNet-50 trained from scratch,” which is later included in fully fine-tuning experiments. It is unclear how R50-Rand is trained prior to evaluation if no distillation is applied, and how it differs from the fully fine-tuned models. Moreover, if R50-Rand is randomly initialized in Table 1, its relatively strong performance compared to other models is surprising and insufficiently explained.

6. The paper does not clearly specify the experimental details for linear probing on frozen encoders versus full fine-tuning. In particular, it is unclear whether decoders or projectors are initialized from scratch during full fine-tuning, and whether the only difference between the two settings is whether the encoder is frozen.

**Detailed Comments:**

None

**Justification Of Final Rating:**

The rebuttal provides a detailed and well-reasoned response to the concerns raised in my review. The authors’ clarifications and additional explanations adequately address my questions and improve confidence in the soundness of the proposed approach and evaluation. Based on these responses, my concerns have been sufficiently resolved.

**Justification Of The Preliminary Rating:**

The paper presents a domain-specific distillation strategy for transferring DINOv3 representations to ultrasound data, with results that indicate meaningful performance gains. While there remain open questions regarding deployment constraints and experimental clarity, the approach is broadly applicable and shows promise for future extensions. On balance, I recommend an accept.

**Questions To Address In The Rebuttal:**

1. The authors should provide a more detailed and evidence-based discussion of deployment constraints in POCUS, including realistic memory, latency, or hardware limits, and clarify whether the proposed compression meaningfully resolves these issues.

2. Reporting end-to-end inference latency, memory usage, or throughput, along with clearer hardware specifications, would help quantify the practical contribution of the distillation framework.

3. The authors should better justify this design choice and include ablation studies to assess whether mixup is necessary and how it affects downstream performance.

4. A clearer and more explicit description of how each dataset is used across distillation, linear probing, and downstream tasks would improve transparency and reproducibility.

5. The authors should clearly define how R50-Rand is trained and how it differs from other baselines across experimental settings.

6. Providing a concise but explicit description of these training protocols would help readers better interpret the reported results.

7. The reviewer encourages the authors to release the code, data, and model weights upon acceptance.

---

> ### Author Response · Authors · 2026-01-25
>
> We sincerely thank Reviewer rGok for the constructive feedback and recognition of our work's practical relevance. We address each concern below:
>
> Deployment Constraints & Practical Benefits:
> We appreciate this crucial point. We added Section 5.4 (Deployment Feasibility for Point-of-Care Ultrasound) and Table 5 with comprehensive benchmarks under a standardized protocol (batch size 1, 224×224, FP32). Our distilled student achieves:
>
> 12.75× faster CPU inference (430.51ms → 33.76ms)
> 12.02× higher CPU throughput (2.30 → 27.64 img/s)
> 45.8% reduction in peak GPU memory (686.70MB → 372.17MB)
> 53.18% fewer FLOPs (22.80 → 10.67 GFLOPs)
> 62.81% fewer parameters while retaining 99.2% of teacher's Dice
>
> These improvements are critical for tablet-based POCUS systems (e.g., EchoNous Kosmos on iPad Air, Philips Lumify on Android) with 4-12GB system memory.
>
> Mixup Justification & Ablation:
>
> We added Section 5.6 and Table 6 with a full 2×2 ablation (±Mixup × Default/US-aware). Key finding: US-aware without mixup achieves BUSI Acc 0.8790, F1 0.8738, outperforming Default+Mixup (0.8712, 0.8602). This confirms US-aware augmentations are the primary driver; mixup acts as a regularizer applied only during distillation.
>
> Data Usage Clarity:
>
> Table 1 now lists all 35 datasets with anatomy labels and references. Section 4.1 explicitly states DDTI and BUSI are held-out downstream benchmarks not included in distillation.
>
> R50-Rand Definition:
>
> Clarified in Section 4.2: "R50-Rand, where weights are randomly initialized at the start of downstream training (no pretraining) and the model is then trained end-to-end using the same downstream protocol as all baselines."
> Training Protocol Details:
>
> Fine-tuning Protocol:
>
> Section 4.3 provides unambiguous specifications:
> Linear probing: Frozen backbone, learned head
> Full fine-tuning: All layers updated with single learning rate and cosine annealing
> No partial unfreezing or layer-wise LR decay
>
>
> We commit to releasing all code and model weights upon acceptance.

---

> > ### Comment · Reviewer_rGok · 2026-02-01
> >
> > The authors provide a detailed rebuttal that adequately addresses my concerns.

---

### Author Rebuttal · Authors · 2026-01-25

**Rebuttal:**

We thank all reviewers for their constructive feedback. We substantially revised the manuscript to improve deployment evidence, attribution, protocol clarity, and diagnostic insight. All the changes has been highlighted.

**NEW: Deployment Feasibility (Section 5.4, Table 5)**
Comprehensive benchmarks: 12.75× faster CPU inference (430→34ms), 45.8% less GPU memory (687→372MB), 53.18% fewer FLOPs, 62.81% fewer parameters while retaining 99.2% of teacher's Dice critical for tablet-based POCUS systems.

**NEW: Augmentation & Mixup Ablation (Section 5.6, Table 6)**
Full 2×2 ablation (±Mixup × Default/US-aware) shows US-aware augmentations are the primary driver (BUSI Acc 0.8790 vs 0.8712 for Default+Mixup). Mixup acts as a regularizer applied only during distillation.

**NEW: Architecture Generalization (Section 5.7, Table 7)**
Distilled ResNet-18 (+9.6% BUSI Acc, +8.9% DDTI Dice) and ConvNeXt-Tiny (+60% BUSI Acc, +20.6% DDTI Dice) validate framework generalizability.

**NEW: FM Comparison (Section 5.8, Table 8)**
R50-Distill-US (25M params, Dice=0.7872) nearly matches DINOv3 (86M, 0.7933) and approaches SAM2-based Temporal (81M, 0.8041).

**CLARIFIED: Data Usage (Table 1)**
Lists all 35 datasets with anatomy labels. DDTI/BUSI are held-out downstream benchmarks.

**CLARIFIED: Training Protocols (Section 4.3)**
Linear probing: frozen encoder, head only. Full fine-tuning: all weights updated. Decoders initialized from scratch. Single LR, matched schedules. Grayscale→3-channel via replication.

**CLARIFIED: DDTI vs. BUSI Gap (Supplementary A)**
Qualitative analysis (Figs. A1-A3): BUSI challenges include lesion variability, normal-class false positives, acquisition heterogeneity, and DINOv3 overfitting.

**Commitment:** Code and weights released upon acceptance.

We believe these revisions comprehensively address all concerns and strengthen our contribution.

**Supporting Material:**

/attachment/d146de136b340610a8d1e1eba770e240216ca9f1.pdf

---

### Meta-Review · Area_Chair_oQxF · 2026-02-12

**Recommendation:** Accept (Poster)
**Confidence:** 4

**Metareview:**

Positive reviews from all three reviewers.

---

### Decision · Program_Chairs · 2026-02-13

Accept (Poster)